# Differential Expression of Stress Adaptation Genes in a Diatom *Ulnaria acus* under Different Culture Conditions

**DOI:** 10.3390/ijms25042314

**Published:** 2024-02-15

**Authors:** Elvira Bayramova, Darya Petrova, Artyom Marchenkov, Alexey Morozov, Yuri Galachyants, Yulia Zakharova, Yekaterina Bedoshvili, Yelena Likhoshway

**Affiliations:** Limnological Institute, Siberian Branch of the Russian Academy of Sciences, 664033 Irkutsk, Russia; bairamovaelvira@gmail.com (E.B.); daryapetr@gmail.com (D.P.); marchenkov.am@gmail.com (A.M.); alexeymorozov1991@gmail.com (A.M.); yuragal@gmail.com (Y.G.); julia.zakharova@gmail.com (Y.Z.); yel@lin.irk.ru (Y.L.)

**Keywords:** diatoms, metacaspases, death-specific protein, delta-1-pyrroline-5-carboxylate dehydrogenase, glutathione synthetases, genome, qRT-PCR, Si starvation, long-term cultivation, bacterial algicidal effect

## Abstract

Diatoms are a group of unicellular eukaryotes that are essential primary producers in aquatic ecosystems. The dynamic nature of their habitat necessitates a quick and specific response to various stresses. However, the molecular mechanisms of their physiological adaptations are still underexplored. In this work, we study the response of the cosmopolitan freshwater diatom *Ulnaria acus* (Bacillariophyceae, Fragilariophycidae, Licmophorales, Ulnariaceae, *Ulnaria*) in relation to a range of stress factors, namely silica deficiency, prolonged cultivation, and interaction with an algicidal bacterium. Fluorescent staining and light microscopy were used to determine the physiological state of cells under these stresses. To explore molecular reactions, we studied the genes involved in the stress response—type III metacaspase (MC), metacaspase-like proteases (MCP), death-specific protein (DSP), delta-1-pyrroline-5-carboxylate dehydrogenase (ALDH12), and glutathione synthetase (GSHS). We have described the structure of these genes, analyzed the predicted amino acid sequences, and measured their expression dynamics in vitro using qRT-PCR. We demonstrated that the expression of UaMC1, UaMC3, and UaDSP increased during the first five days of silicon starvation. On the seventh day, it was replaced with the expression of UaMC2, UaGSHS, and UaALDH. After 45 days of culture, cells stopped growing, and the expression of UaMC1, UaMC2, UaGSHS, and UaDSP increased. Exposure to an algicidal bacterial filtrate induced a higher expression of UaMC1 and UaGSHS. Thus, we can conclude that these proteins are involved in diatoms’ adaptions to environmental changes. Further, these data show that the molecular adaptation mechanisms in diatoms depend on the nature and exposure duration of a stress factor.

## 1. Introduction

Diatoms (Bacillariophyta) are a major component of phytoplankton and heavily contribute to the primary production of carbon [1] and the silicon biogeochemical cycle [2]. As diatoms are an essential part of water ecosystems, it is very important to understand their population control mechanisms. Recently, it was shown that cell reactions of microalgae in natural populations of phytoplankton are more varied than was supposed earlier. Data shows that phytoplankton species evolved molecular systems of stress adaptation, signaling, and PCD launching [3,4,5]. Diatoms coexist and interact with other organisms in aquatic ecosystems, such as consumers [6], bacteria [7], and viruses [8], competing with them for nutrients [9] and light [10].

All living cells show some reaction to stressful environments, usually by either activating adaptation mechanisms or dying due to the breakdown of intracellular processes [11]. These adaptation mechanisms can involve compensation for deficient nutrients [12] and the elimination of toxic effects caused by oxidative stress [13] or disorders in protein folding [14] or DNA structure [15]. In addition, cells under negative environmental influences may pass around signaling molecules that induce metabolic changes in target cells allowing them to escape negative consequences [16] or initialize cell death [17].

Important component of biochemical cell homeostasis is the redox potential, allowing the antioxidant defense system (ADS) to eliminate extra reactive oxygen species (ROS) via enzymes capable of neutralizing ROS and reducing oxidized molecules [18]. If a cell is unable to maintain its inner homeostasis under an intense stress, ROS accumulate and the cell undergoes oxidative stress, i.e., oxidized biomolecules disrupt DNA and protein structures and break the integrity of membranes, leading to cell death, including programmed cell death (PCD), which is caused by specific cellular processes [11,19]. PCD has explicit morphological manifestations such as the degradation of organelles, vacuolization, DNA fragmentation, and phosphatidylserine externalization [20]. The process is induced by the activation of specific cysteine proteases, such as caspases, that invoke a specific hydrolytic degradation of certain proteins after aspartate residue [21]. Caspases are also involved in the cell cycle and cell differentiation, as well as in cytoskeleton reorganization [22,23,24]. Metacaspases, homologues of animal caspases, were found via bioinformatics analysis in plants, fungi, and protists [25]. 

Due to the important role that animal caspases play in both normal cell functioning and cell death, an in silico search for their homologs was performed on other organisms. A similar family of proteins was discovered in protists, fungi, and plants and dubbed metacaspases [25]. There is a high degree of homology between caspases and metacaspases, but their substrate specificity differs as metacaspases perform proteolytic cleavage of proteins after aspartate or lysine [26]. Despite the difference in specificity, there is evidence of a relationship between metacaspases and cell death in various organisms, including embryophytes [27,28], yeasts [29], protozoa [30], cyanobacteria [3], and microalgae [31]. Three types of metacaspases are distinguished by their domain structure: type I, typical of many groups, including proteobacteria, fungi, and plants, contains two domains (p10 and p20); type II, found only in plants and algae, has additional domains; and type III only occurs in unicellular algae that underwent secondary endosymbiosis [31,32]. Type III differs from the other types in the rearrangement of the small (p10) and the large (p20) domains. In addition, metacaspase-like proteases (MCPs) containing only the p20 domain have been found in algae, bacteria, and fungi [31,32]. MCPs lack the small p10 domain important for the calcium-dependent activation of metacaspases [33,34]. This raises questions about their functional activity and substrate specificity [31].

Type III MCs and MCPs are typical of diatoms [31,32]. Constitutive MC transcription observed in *Thalassiosira pseudonana* suggests these proteases are involved in normal cell activities [35]. Diatom MCs become active in *T. pseudonana* under iron starvation and long-term cultivation [35]. Silicon [36], phosphorus, and nitrogen [37] starvation also activate MCs in *Skeletonema marinoi*. In the process of culture aging or under nutrient deficiency, diatoms produce signaling molecules such as polyunsaturated aldehydes (PUAs) that penetrate other cells of the population and activate a mechanism leading to their death [38,39]. A study of one PUA variant (decadienal) revealed an MC activation in *Phaeodactylum tricornutum* [40,41]. These results suggest that metacaspases may be involved in controlling diatoms’ population size.

Another protein that is potentially important for understanding the stress response of diatom cells is death-specific protein (DSP). A study of *Skeletonema costatum* cells in their stationary growth phase revealed nuclear DNA fragmentation and increased expression of death-specific protein (DSP) [42]. Further studies showed the presence and increased transcription of this protein in other diatom and haptophyte species during the stationary growth phase [37,43,44]. This protein is homologous to the PGR5 (thylakoid-associated proton gradient regulator-5) protein [45] involved in the regulation of cyclic electron flow around photosystem I (PSI) in land plants [46]. Data show that TpDSP1 overexpression induced by iron starvation enhances cyclic electron flow around PSI in *T. pseudonana*, thereby leading to increased ATP production and promoting the adaptation and growth of cells [45]. Experiments with another diatom, *P. tricornutum*, showed DSP involvement in the transcription regulation of photosynthesis-related genes, protecting cells against oxidative stress under iron starvation and light deficiency [44]. Increased DSP expression was observed in *Skeletonema tropicum* [47] under the influence of PUAs (octadienal) and hydrogen peroxides. A key feature of DSP is the presence of two calcium-binding EF-hand domains, suggesting its involvement in calcium-dependent signaling [42,48].

Like other organisms, diatoms have a multicomponent antioxidant defense system that eliminates ROS and reduces oxidized elements [18]. The glutathione, involved in reducing oxidized proteins, is an important ADS component [49]. Glutathione requirement of a cell may be studied by analyzing glutathione synthetase (GSHS) expression [50,51]. The cell redox potential is supported with proline, which is capable of binding ROS and functioning as a molecular chaperone [52]. The delta-1-pyrroline-5-carboxylate dehydrogenase enzyme (ALDH12) from the aldehyde dehydrogenase superfamily plays an essential role in the proline cycle [53]. ALDH12 has been found in bacteria, unicellular eukaryotes (e.g., heterokonts), plants, and animals [54]. It participates in *Glycine max*’s reaction to drought [55], *Zea mays’* and *Physcomitrella patens*’ response to salinity stress [54], and diatoms’ reaction to a nutrient deficiency, e.g., silicon [36,37].

The silicon concentration in the environment is a limiting factor for the growth of diatoms that use it for their siliceous frustules [56]. Silicon starvation induces disorders in the mitochondrial respiratory chain and chloroplast photosynthetic chain, provoking oxidative stress [57,58] and inhibiting cell division in diatoms [59,60]. For example, in *Skeletonema marinoi*, the transcription of MC and ADS genes increases on the second day of silicon starvation, followed by the progressive development of PCD-relevant processes [36]. Similar morphological and molecular signs were described for the stationary growth phase in the culture of the diatoms *S. costatum* and *T. pseudonana* [35,42]. The stationary growth phase leads both to the depletion of nutrients and to the accumulation of secondary metabolites, including signaling PUAs [38,39,47].

Interactions between diatoms and bacteria can sometimes be mutualistic [61,62], but this process can also involve diatoms competing with bacteria for nutrients [63], adapting their metabolism [64], and producing antibacterial substances [65]. Bacteria similarly produce algicidal agents, provoking stress in diatoms [66,67]. Algicidal bacteria are commonly used in limiting the toxic algal blooms produced by diatoms [63,67], but the cellular defense mechanisms that diatoms deploy against these bacteria have not been thoroughly studied. We have previously shown that a *Bacillus mycoides* strain BS2-15 isolated from Lake Baikal sediments had an algicidal effect on the diatom *Ulnaria acus* (Kützing) Aboal (Bacillariophyceae, Fragilariophycidae, Licmophorales, Ulnariaceae, *Ulnaria*). Under the influence of this bacterium, the diatom culture stopped growing during the first three days of co-culture, followed by rapid death during the next seven days, during which the accumulation of neutral lipids and PCD features such as nuclear membrane injury and DNA fragmentation were observed [68]. Thus, one of the goals of this work was to study the response of *U. acus* with regard to the algicidal effects of this bacterium.

Most of the diatom stress response research has focused on marine species [69]; we, on the other hand, have chosen *U. acus* as a model object. It is a cosmopolite freshwater araphid diatom, the only such diatom to have an annotated genome [70] and a transcriptome focused on differential expression under changing light conditions [71]. A previous analysis of this species revealed eight contigs in which metacaspase-coding genes were identified [72]. *U. acus* is a dominant plankton species in the deepest freshwater lake on the planet, Lake Baikal, and thus one of the major primary producers in the lake’s ecosystem [73,74]. Long-term studies of phytoplankton dynamics suggest that *U. acus* outcompetes other spring phytoplankton species [75], but it is unclear what exactly gives it an advantage.

The main goal of this work was to describe several genes involved in *U. acus*’ stress response (*MC*, *MCP*, *DSP*, *ALDH12,* and *GSHS*) and measure their expression under non-lethal stresses. Results for these stresses may generalize to other unexpected environmental changes that diatoms face in the wild.

## 2. Results

To study the expression of genes involved in the diatom stress response, we first needed to detect them in *U. acus* genomic data, confirm the sequences with Sanger sequencing, and check whether functionally important sites and motifs remained intact.

### 2.1. MC and MCP Molecular Structure

The length of *U. acus* metacaspase genes varied between 957 (*UaMC1*) and 2106 bp (*UaMC4*, *UaMC5*) (Appendix A). FGENESH intron–exon structure analysis revealed two introns in the sequence of *UaMC2* and one intron in the sequence of *UaMCP6*–*UaMCP8*, while no introns were detected in sequences of *UaMC1*, *UaMC3*, *UaMCP4*, and *UaMCP5* (Appendix A).

The amino acid sequences encoded by related genes were predicted to be 306 to 701 a.a. long. The homology of predicted amino acid UaMC1–UaMC3 sequences was not high (32–46%); their homology to other *U. acus* MC sequences did not exceed 20% (Appendix A). Two specific caspase domains (p10 and p20) typical of the C14 cysteine peptidase family were identified in the UaMC1–UaMC3 structure (Table 1); the p10 domain preceded the p20 domain in these sequences. Analysis of specific amino acid motifs required for the functional activity of metacaspases revealed that the structure of the p10 domain of these proteins contains a D(X)QTSAD motif typical of metacaspases (Figure 1A). However, aspartic acid residue (D), necessary for binding calcium ions, was replaced with serine (S) in UaMC3. It has been reported that the p20 domain structure contains a catalytic C-H (cysteine–histidine) dyad in SGHG and DCCH, forming a protease active center [15,72]. According to the analysis, this group preserved these motifs as well as the calcium-binding site (DLP tripeptide) (Figure 1A). Thus, UaMC1–UaMC3 are type III metacaspases according to their main characteristics (Table 1).

The UaMCP4 and UaMCP5 sequences were 98.7% homologous, each being 701 a.a. long (Appendix A). The specific caspase p10 domain was not identified in the sequences (Table 1). The p20 domain included DLP tripeptide and a fully preserved SGHG motif. However, positively charged histidine in both sequences of the DCCH motif was replaced with negatively charged D (Figure 1A).

The predicted UaMCP6–UaMCP7 amino acid sequences were identical and 97% homologous to the UaMCP8 sequence. These MCPs typically only have the p20 domain whose structure preserves the calcium-binding site (DLP), but do not have SGHG and DCCH motifs forming an active center (Figure 1A).

Phylogenetic reconstruction revealed a segregation of MC and MCP sequences between two clades. UaMC1-3 formed a clade with the type III metacaspases of *P. tricornutum* and *T. pseudonana*, while UaMCP4-8 formed a clade with the same diatoms’ MCP (Figure 1B).

To gain insight into the localization of these proteins inside the cell, we analyzed the predicted amino acid sequences in DeepLoc 2.0, which uses neural networks to estimate protein localization based on a training set of experimentally localized proteins on UniProt [76]. This algorithm predicted a cytosolic localization for all UaMCs and UaMCPs (Figure 1C).

### 2.2. Molecular Structure of UaDSP

The *UaDSP* gene located in contig 2664 was detected by scanning the predicted amino acid sequences of the *U. acus* genome; for this purpose, we used hmmsearch with the hidden Markov model of the EF-hand domain (PFAM ID PF13499) (as a query) on published DSP gene sequences of eight microalgae (Appendix A). *UaDSP* is 645 bp long (GenBank No NCBI–OR231940). According to FGENESH analysis, the gene has no intron and encodes a protein 214 a.a. long (Appendix A). The analysis of the predicted amino acid sequence showed the presence of a signaling peptide at the N-terminal protein part, typical of most DSPs (Appendix A; Appendix A). Two EF-hand domains were identified at the C-terminal protein part (Figure 2; Appendix A). The analysis of the DxDxDG motif in the structure of the domains demonstrated that one D was replaced with S in the first EF-hand domain of UaDSP, while the motif was completely preserved in the second domain. Other diatoms show replacements in the DxDxDG structure in both domains. DeepTMHMM revealed the absence of transmembrane domains in UaDSP (Appendix A). NetPhos-3.1 detected 17 phosphorylation sites in the predicted UaDSP amino acid sequence (Appendix A). The DeepLoc 2.0 algorithm points at an extra-cellular localization of the protein (Figure 1C).

### 2.3. Molecular Structure of UaALDH12

A single 1783 bp long delta-1-pyrroline-5-carboxylate dehydrogenase UaALDH12 gene GenBank access No NCBI—OR231941) was detected in contig 22733 by means of hmmsearch using the hidden Markov model of the ALDH domain (PF00171) as a query and diatom ALDH12 sequences published in NCBI (Appendix A). According to the FGENESH analysis of gene structure, an intron 100 bp long begins at nucleotide position 575. The analysis of the predicted amino acid sequence (560 a.a.) shows that it is a member of ALDH-SF (the aldehyde dehydrogenase superfamily), the most similar to delta-1-pyrroline-5-carboxylate dehydrogenase (ALDH12). The identity of UaALDH12 to AtALDH12 (*A. thaliana*) was 53.9%, reaching 70% compared to homologous ALDH12 sequences in other diatoms (Appendix A). At the same time, there was variability in the area of the last 80 a.a. at the N-terminus of the protein (Appendix A). Comparative analysis of the predicted amino acid sequence revealed that nonpolar valine (V) in the NAD^+^-binding FTGSSV site was replaced with polar positively charged arginine (R); such replacement is typical of both ALDH12 in diatoms and AtALDH12 in *A. thaliana* (Figure 3). In the catalytic VKLEDAG site, nonpolar leucine (L) was replaced with other nonpolar amino acid residues such as valine (V) in the UaALDH12 sequence or isoleucine (I) in other diatom sequences (Figure 3). Glutamine (E) and cysteine (C) active sites typical of enzymes of this group were also found (Figure 3). The DeepLoc 2.0 algorithm predicted a localization of the protein in mitochondria (Figure 1C).

### 2.4. Molecular Structure of UaGSHS

A single *GSHS* gene 1497 bp long (GenBank access No OR231942) was detected in contig 5207 of the *U. acus* genome by means of the hidden Markov glutathione synthetase model (GSHS, PF03199), as a query using GSHS amino acid sequences in GenBank NCBI. According to FGENESH, the gene has no introns and encodes a protein 498 a.a. long (GenBank access No OR677821). Since GSHS has not been previously studied in diatoms, we had no choice but to compare the UaGSHS sequence with homologs from model eukaryotes. The predicted UaGSHS protein sequence is 33% identical to the GSHS of *Homo sapiens* (HsGSHS) and *A. thaliana* (AtGSHS) (Figure 4). In the UaGSHS sequence, we found an active center formed by the γ-glutamylcysteine/glutathione-binding site and ATP-binding site with a single replacement in the structure of glutamylcysteine/glutathione-binding site where serine (S) is located at a position homologous to the S153 sequence of *H. sapiens* (Figure 4). UaGSHS sequence alignment showed that the substrate-binding loop (S-loop) and alanine-rich loop (A-loop) are not conserved, while the glycine-rich loop is completely preserved (Figure 4; Appendix A). The DeepLoc 2.0 algorithm was used for the analysis of the predicted amino acid sequence points at a cytosolic localization of the protein (Figure 1C).

### 2.5. Differential Expression of Six Studied Genes as a Response to Different Stresses

We performed experiments with an axenic *U. acus* culture under three different stresses: the absence of silicon in the medium, long-term cultivation, and exposure to algicidal bacterial filtrate. Our goal was to study the expressions of *UaMC1*, *UaMC2*, *UaMC3*, *GSHS*, *ALDH12,* and *DSP* under non-lethal conditions, when the cells are still responding to external stresses. To estimate the severity of stress, we observed whether the cells stopped dividing, and measured lethality using vital fluorescent staining (Figure 5A).

Si Starvation. The quantity of *U. acus* cells cultivated in DM medium was increased throughout the experiment (Figure 5B). Cells stopped dividing in silicon-deficient conditions and the number of cells remained constant, with 80% (±5%) of cells being alive. qPCR gene expression analysis (Figure 5B) showed that the expression of *UaMC1* in the silicon-starved cells exceeded reference values on the third day of culture, reached its maximum on the fifth day, and decreased to the reference values on the seventh day. The expression of *UaMC2* in the silicon-deficient medium increased insignificantly, reaching a significant difference with the reference values on the seventh day. The expression of *UaMC3* increased on the third day in the silicon-deficient medium and then fell insignificantly. On the seventh day of Si starvation, the expression of ADS genes increased, with *ALDH12* expression being higher than that of *GSHS*. The *DSP* expression level was increased from day 1 to day 5 of Si starvation; after that, it fell to the reference values.

Prolonged Culture. *U. acus* cells were inoculated into a fresh DM medium and cultured for 60 days without refreshing the medium. Cells grew for the first 30 days (Figure 5C) and then stopped. However, according to cell staining using a live cell labeling kit, around 80% (±10%) of cells were alive. On day 45, the expression of *UaMC1* and *UaMC2* increased, peaking on day 60 (Figure 5C). *UaMC3* expression was low throughout the experiment. *GSHS* expression fell on day 30, but increased during the stationary growth phase, and reaching its maximum on day 60, which was lower than that of *UaMC1* and *UaMC2*. *ALDH12* expression fluctuated insignificantly during the experiment, while that of *DSP* gradually increased during the entire period; however, its values, like *GSHS*, were lower than those of metacaspases.

Exposure to Algicidal Bacterial Filtrate. We counted the number of *U. acus* cells in the presence of the algicidal *B. mycoides* bacterial filtrate for five days. *U. acus* cells inoculated into the medium with the bacterial filtrate stopped dividing (Figure 5D). The expression of *UaMC1* under the effect of the algicidal filtrate was already comparable to the reference values on day 2 and continued increasing on day 3. However, its values remained low compared to *GSHS*, whose expression gradually rose during the entire period, showing the most manifest reaction in the experiment. The expression of *UaMC2* was comparable with the reference values throughout the experiment. The expressions of *UaMC3*, *ALDH12*, and *DSP* in the experiment fluctuated within the static error range and did not exceed reference values.

The experiments show that the expression of all genes changed differentially depending on the type and duration of the stressor.

## 3. Discussion

This study aimed to further our understanding of MC, DSP, GSHS, and ALDH12’s roles in the *U. acus* stress response. Metacaspases are known to take part in diatom cell death in stressful environments [35,36]. Works on DSP point to its role in arresting diatom division as well as in adapting to changing light conditions [42,43,45]. GSHS and ALDH12, being components of the antioxidant system, protect cells from oxidative stress induced by negative environmental factors [36,37]. Diatom blooms are controlled by multiple biotic and abiotic factors, including algicidal bacteria [63], and the shortage of nutrients, including silica [77]. Our work studied the changes in the expression of these genes during active cell proliferation and when division stopped during prolonged culturing. We also tracked the effect of a silica shortage and an algicidal strain of *B. mycoides* previously shown to kill *U. acus* cultures [68].

### 3.1. Comparison of the Molecular Structure of the Predicted U. acus Proteins with Their Homologues in Other Organisms 

MCs and MCPs. Here, we studied metacaspases as potential participants in the molecular response to stress in culture conditions. We were also able to separate the genes encoding MCs and MCPs (Table 1). It is noteworthy that the *UaMC1* gene had already been described as encoding a duplicated amino acid sequence [72]. Since we did not find duplicated elements (Table 1), we may consider this gene to be encoding a type III metacaspase representative. 

The analysis of the predicted amino acid sequences shows that the proteins produced by these genes should be functional. A small p10 domain preceding a large p20 domain was found only in UaMC1–UaMC3 (Figure 1A, Table 1). Such a domain architecture, different from type I and type II metacaspases, was previously described in *T. pseudonana* and *P. tricornutum*. It is a typical feature of type III metacaspases [31]. The p10 domain is characterized by a D(X)QTSAD sequence, where aspartic acid (D) residues are required for binding calcium ions, which is important for the regulation of metacaspase activity [33,34]. The UaMC3 motif looks like SEQTSAD (Figure 1A). It is known to lose functional activity if both aspartic acid residues (D) are missing [33,34]; hence, we suggest that the replacement of the first amino residue D should not disrupt protein activity and the ability to participate in biochemical processes.

The histidine (H) and cysteine (C) dyad (in SGHG and DCCH motifs, respectively) form a protease active center known to be located in the p20 domain [15,72]. Glycine (G) residues in the SGHG site are also supposed to play a stabilizing role in the structure of active centers [26]. These motifs were not completely preserved in the UaMCP4–UaMCP8 sequences, bearing witness to disorders in the formation of active centers in the structure of these proteins. Thus, UaMCP4–UaMCP8 are not only MCPs rather than MCs, as they only contain a p20 domain; they also probably lack protease activity due to the disruption of the active center structure. 

The MCP domain structure in phytoplankton specimens has been described [31], but the function of these proteins in a cell was not assumed. It is known that the expression of MCPs in certain experiments was lower than that of metacaspases [32,41].

Since we found only three (*UaMC1-UaMC3*) genes in the *U. acus* genome encoding proteins that could correspond to type III metacaspases, according to their structural characteristics, we focused our further investigations on understanding the cell response, and the function of the expression products of these genes is responding to stress conditions in culture.

DSP. At the present time, DSP has been identified only in the genomes of diatoms [43,46] and coccolithophores (*E. huxleyi*) [48]. It is noteworthy that both centric diatoms and haptophytes have genes encoding two isoforms of this protein. We identified a single gene encoding UaDSP in the genome of *U. acus*. ScDSP1, PtDSP, and TpDSP1 were shown to be localized inside plastids and involved in electron transfer and carbon fixation [44,48], while EhDSP and FcDSP did not reveal such localizations. The predicted transmembrane domain found in ScDSP1 and TpDSP1 is suggested to participate in peptide fixation in the thylakoid membrane [42,45]. Analysis of the predicted UaDSP amino acid sequence showed neither the presence of the transmembrane domain nor its potential localization inside plastids; similar data were obtained for DSP in other diatoms.

DSP contains two EF-hand domains typical of proteins involved in calcium-dependent signaling [78]. The EF-hand domain is characterized by a DxDxDG motif participating in Ca^2+^ binding [79]. These motifs are present in both EF-hand domains of the *U. acus* DSP structure, but only that of the second EF-hand domain is fully conserved (Figure 2). It should be noted that DxDxDG motifs are also not conserved in the other predicted DSP amino acid sequences in diatoms, but were proved to be functional [42,44,45]. Analysis of post-translational modification sites in the predicted UaDSP amino acid sequence revealed the presence of 17 putative phosphorylation sites. However, all three sequences were found to preserve the predicted phosphorylation sites in only three homologous positions (Appendix A) compared to TpDSP1 in *T. pseudonana* [45] and PtDSP in *P. tricornutum* [44]. The conservation of phosphorylation sites in the diatom sequences of miscellaneous systematic groups may indirectly confirm the importance of protein functionality regulation via its post-translational modification.

The ubiquity of DSP-like sequences in ecological metagenome and metatranscriptome datasets emphasizes the significance of this protein in physiological processes taking place in phytoplankton populations under different conditions [45].

*ALDH12.* Even though representatives of the aldehyde dehydrogenase multifamily are important enzymes involved in the adaptations to stress factors, this group of enzymes remains unstudied in diatoms [53]. One of the members of this multifamily in plant organisms, inherent both in their higher and lower representatives, is delta-1-pyrroline-5-carboxylate dehydrogenase [54], which is involved in proline homeostasis in cells and important in the stress response [80,81].

A single gene coding an amino acid sequence highly homologous to the ALDH12 sequence of a model plant organism, *A. thaliana* (Figure 3), was found in the *U. acus* genome. An intron–exon organization has been shown for this gene in *A. thaliana* [54]. According to *UaALDH12* sequence analysis, there is one long intron, confirming the well-known fact that diatom genomes have fewer introns than plants [82]. The high homology with the ALDH12 of higher plants (Figure 3) shows that UaALDH12 belongs to the same group of aldehyde dehydrogenases and can participate in the cell in response to oxidative stress.

*GSHS.* Analysis of the predicted *U. acus* glutathione synthetase amino acid sequence in comparison to the GSHS of other model organisms (*H. sapiens* and *A. thaliana*) showed the presence of loops required for the access to the enzyme active center [83], as well as γ-glutamylcysteine/glutathione-binding and ATP-binding sites forming the protein active center [84] (Figure 4). This allowed us to assume that the *U. acus* GSHS plays a key role in cell defense against oxidative stresses and adaptations to changing environments.

### 3.2. Changes in MC, ALDH12, GSHS, and DSP Expression

A relevant way to study the molecular mechanisms activated in a cell under various conditions is qPCR, which reflects changes in the expression of genes of interest [85]. Since metacaspases are assumed to be involved in various processes and functions, a selective study of each individual metacaspase is required [35,36,37]. At present, determining the expression level of target genes using qPCR is probably the most accurate method of detecting changes in the activity of certain proteins, even though the onset of their activity may be delayed.

Previously, we obtained the *U. acus* transcriptome with quantitative data that allowed us to determine different levels of expression of target genes under the exponential growth phase and normal light, during 48 h in darkness, and under light after synchronization in darkness (20 and 40 min after return to light) [71]. According to the reported data, the expression of *GSHS* and *ALDH12* did not change in any conditions, confirming the lack of stress in those experiments. *UaMC1* had the highest expression level among the three metacaspases, both under the exponential growth phase and synchronization in the dark. *UaMC1* and *UaMC2* transcription levels decreased significantly in the dark, while only *UaMC3* expression decreased after the return of light (Appendix A). According to transcriptome analysis data [71], the *DSP* transcription level increased when the culture was placed in the dark. This confirms the role of DSP in light adaptation processes [44,45]. Nonetheless, the return of dark-acclimated cells to light did not affect protein transcription (Appendix A).

To study the MC, DSP, ALDH, and GSHS involvement in the physiological adaptation of *U. acus* to stresses, we cultured cells under Si starvation as a key division-limiting factor, in conditions of a prolonged culture where cells are known to stop division as a result of depletion in nutrients [86], and under exposure to an algicidal bacterial filtrate. We had previously demonstrated that *U. acus* cells stopped dividing during the first three days and perished rapidly during the next seven days after the addition of the *Bacillus mycoides* bacterial filtrate [68]. We suggested that the bacteria-produced algicidal agent activated mechanisms against stress. Therefore, we analyzed the expression of the studied genes in the first three days of culture with a bacterial filtrate.

In all experiments, the expression of *UaMC1* changed most significantly. It should be noted that according to our phylogenetic reconstruction, a marine diatom *P. tricornutum*, under nitrogen starvation and light regime variations at different concentrations of iron, demonstrated an active expression of PtMC5 [41,87,88], forming a unified subclade with UaMC1. It is important to note a higher level of *UaMC1* expression compared to other studied metacaspases (Appendix A) according to the published *U. acus* transcriptome [71]. These metacaspases are located on the phylogenetic tree close to TpMC1 (Figure 1B), which showed decreasing expression levels in *T. pseudonana* under iron starvation [35]. Our results reveal that diatom metacaspases could be related to the stress response invoked by the deficiency of nutrients such as silicon.

The expression of *UaMC2* was highest at the stage of culture growth arrest. It also slightly increased on the seventh day in the silicon-deficient medium. On the phylogenetic reconstruction, this metacaspase borders with metacaspase-like proteases such as TpMCP5-6 and TpMC3, whose expression in *T. pseudonana* decreases under iron starvation [35]. However, the PtMC2 of *P. tricornutum* from the same clade showed a higher expression under nitrogen deficiency [41,88]. These metacaspases are likely involved in cell response processes under the deficiency of some environmental nutrients.

*UaMC3* demonstrated the least obvious response to the stress conditions studied. Its expression level increased only on the third day of silicon starvation, falling to the reference values on the seventh day. In the phylogenetic reconstruction, UaMC3 is located close to PtMC4, whose gene expression is stable in a complete medium and nitrogen deficiency but decreases in darkness [41,87,88]. According to transcriptome data [71], *UaMC3* responds to changing light conditions via reduced transcription levels after being returned to light and after a synchronization in darkness. Thus, *UaMC3* is not involved in the response to the selected stress conditions. These data reveal different functions of MCs under the selected conditions, which is also confirmed in experiments with other diatoms [35,36,37,41].

The genes of ADS enzymes (ALDH12 and GSHS) showed a different response to the experimental conditions both in our work and in some other papers [36,37]. The expression of *ALDH12* only starts increasing on the seventh day of culture in a silicon-deprived medium (Figure 5B), and remains almost at the same level in prolonged culture (Figure 5C). In experiments with *S. marinoi*, the expression of *ALDH12* did not change during five-day silicon starvation [36], but became higher on day 4 of nitrogen and phosphorus deficiency [37]. The predicted *U. acus* ALDH12 amino acid sequence was highly identical to the sequences of evolutionarily remote diatoms and a model organism *A. thaliana* (>50%). This indicates the importance of the mechanisms in which this protein is involved in during the stress response. *ALDH12* expression changes most significantly under silicon starvation. At the same time, we noted that although the cells stopped dividing, they were alive. The oxidative stress induced by the longer lifetime of parent cells may be related to the accumulation of proline, which is involved in maintaining cell redox potential [52], as its formation is a mode of cytosol pH regulation [89]. ALDH12 is known to be one of the main components of the proline cycle, oxidizing it to glutamate [53].

The expression of *GSHS* increased under all stresses studied. In experiments with *S. marinoi*, the expression level increased briefly under nitrogen and phosphorus deficiency [37] but remained stable under silicon starvation [36]. These conditions likely activate the synthesis of glutathione as an antioxidant that protects the cell against negative ROS effects [90].

DSP is suggested to be a marker of the termination of diatom blooms since the expression of the gene encoding it increases at the stationary growth phase of *S. costatum* culture [42], as well as at exposure to polyunsaturated aldehydes that are infochemical metabolites of diatoms progressively accumulated in culture [47]. Our study also confirmed the correlation between a higher *DSP* expression level and the stationary growth phase. The example of *S. marinoi* showed that *DSP* expression was lower on day 8 of silicon starvation [43]. In our study, the *DSP* expression level increased during the first days of silicon starvation, but then fell substantially (Figure 5). One may suggest that this protein is involved in the processes taking place under reduced concentrations of nutrients, including silicon, required for cells. According to our data, DSP activation is not related to the influence of toxic substances.

In this work on the molecular mechanisms of *U. acus* response to changing environments, we have identified and documented its *MC*, *MCP*, *DSP*, *ALDH,* and *GSHS* genes. We showed that the genes selected for further analysis retain all motifs necessary for their enzymatic activity.

*U. acus* cells stop dividing under prolonged culture, silica limitation, and treatment with algicidal filtrate. However, fluorescent staining in vivo proves that the cells remain alive and can adapt to these conditions. Gene expression analysis shows that *U. acus* cells starved in silicon experience oxidative stress under all three conditions and respond with an increased expression of *UaGSHS* and *UaALDH12*. Metacaspases, a family of proteins previously studied, mainly in the context of cell death, also seem to be involved in the adaptation to these stresses, as their expression changes even when cell death is not induced. The results suggest that UaMC1, UaMC2, UaDSP, and UaALDH12 are primarily involved in responding to silica deficiency, while UaGSHS is most active in the presence of toxic metabolites (whether produced by algicidal bacteria or other *U. acus* cells in aging culture). We also note that UaMC3 is activated during a silica shortage. To conclude, the studied genes activate in different combinations under different stresses, suggesting that diatom cells possess multiple distinct reactions to stressful environments.

## 4. Materials and Methods

### 4.1. Culture Conditions 

An axenic strain BK497 was obtained from *U. acus* cells isolated in March 2020 from a sample collected under ice near Bolshiye Koty settlement, Lake Baikal, according to a protocol [91]. Cells were cultivated in the DM medium [92] at 4 °C and under light intensity 16 μmol/m^2^/s with a 12 h:12 h, light/dark cycle for two weeks. Axenicity of culture was checked with DAPI under microscope Axiovert 200 (Zeiss, München, Germany) with UV filter.

### 4.2. Target Gene Identification

Genes of metacaspases, delta-1-pyrroline-5-carboxylate dehydrogenase, glutathione synthetase, and death-specific protein were searched in the previously published *Ulnaria acus* (=*Synedra acus* subsp. *radians*) genome [71]. Hmmer software version 3.4. [93] and hidden Markov models from the Pfam data base [94] were used for the search.

### 4.3. Sanger Sequencing and Sequence Analysis

To specify MC nucleotide sequence, we designed primers on different sites of contigs obtained via whole genome sequencing (Appendix A, Appendix A). DNA was isolated according to a protocol [95].

Amplification products were analyzed in 1% agarose gel and were purified with a Monarch DNA Gel Extraction Kit (New England Biolabs, Ipswich, MA, USA). They were sequenced using BigDye Terminator v.3.1 (Applied Biosystems, Foster City, CA, USA) and analyzed on 3130XL or 3500XL genetic analyzer (Applied Biosystems, Foster City, CA, USA) in SB RAS Genomics Core Facility (Novosibirsk, Russia). The nucleotide sequences were edited by means of SnapGene Viewer (Dotmatics, available at snapgene.com); alignment of nucleotide sequences was performed in MEGA 7.0. software with the use of ClustalW algorithm [96]. FGENESH was applied for determining exon–intron structure of genes (by Softberry) [97] (parameter settings: organisms = *P. tricornutum*; suboptimal exon cutoff = 0.1).

### 4.4. Analysis of the Predicted Amino Acid Sequences

MC, MCP, DSP, GSHS, and ALDH12 sequences of diatom algae and model organisms were selected for comparative study form GenBank Database (Table 1, Appendix A; Appendix A). ClustalW algorithm in BioEdit v. 7.2.5 software [98] was applied for alignment and analysis of predicted amino acid sequences. Phylogenetic analysis was performed via method of maximum likelihood estimation in MEGA 7.0. software with 1000 bootstrap replications using LG+G evolutionary model.

MC domains and their linker were determined using SMART service [99]. Transmembrane domains were calculated via DeepTMHMM version 2.0. software [100]. DeepLoc-2.0 [101] was used for predicting protein localization. Signaling peptide was identified by means of SignalP 5.0. [102]. Phosphorylation sites were identified with the use of NetPhos-3.1 [103].

### 4.5. Algicidal Effect

An algicidal *Bacillus mycoides* BS2-15 strain [68] was cultured in 1% peptone water at 25 °C for 48 h until it reached an optical density of 1.0 at A600 nm. A cell-free bacterial filtrate was obtained by filtration of bacterial culture through a sterile 0.20 µm analytical track-etched membrane filters–ATM (Reatrack, Obninsk-3, Russia). The bacterial filtrate was added to *U. acus* cells during the exponential growth phase in concentration 100 µL/mL. Cells were calculated in triplicate in 10 µL of culture after mixing, starting from the first day of the experiment in 24 h intervals. Cells for RNA isolation were collected thrice in 24 h intervals.

### 4.6. Silicon Starvation

For silicon starvation experiments, *U. acus* were grown up to concentrations of 1000 cells/mL, and were then filtered through analytical track-etched membranes filters with 3 µm pores (Reatrack, Obninsk-3, Russia). Cells were transferred into 600 mL of silicon-deprived medium where they were cultivated at 4 °C and normal light for 24 h. Cells were counted in triplicate in 10μL of culture after mixing, beginning from the first day of the experiment during seven days in 24 h interval. For RNA isolation, the culture was sampled three times from the first days of the experiment during seven days in 24 h interval.

### 4.7. Long-Term Cultivation

To study the impact of long-term cultivation on diatoms, axenic *U. acus* cells were kept in culture at normal conditions for 60 days. Cell counting began from the first day of inoculation, while the biomass of RNA isolation was collected after 15, 30, 45, and 60 days from the beginning of cultivation.

### 4.8. Fluorescence Microscopy

To identify the living cells in *U. acus* culture, cells were sampled before and after the experiment and stained with live cell labeling kit (ab176735; Abcam, Cambridge, UK). Stock solution and staining procedure were made according to recommendation of manufacturer. Cells were stained in DM medium at 8 ℃ in during 30 min and examined with the epifluorescence Axiovert 200 microscope (Zeiss, München, Germany) equipped with a blue optical filter (495/515 nm). Stained cells were counted among 100 cells that were randomly selected. Microphotographs were made with the Olympus IX83 (Olympus, Tokyo, Japan) microscope and Olympus DP74 digital camera.

### 4.9. RNA Isolation and cDNA Synthesis0

Biomass was collected on analytical track-etched membrane filters with 3 μm pores (Reatrack, Russia) and resuspended in 1 mL of IntactRNA (Evrogen, Moscow, Russia) according to manufacturer’s recommendations. The samples were centrifuged at 5000 g for 2 min (Eppendorf, 5415R, Hamburg, Germany). Supernatant was removed, and the precipitate was placed into liquid nitrogen and then pounded with a pestle. Total RNA was extracted from homogenized sample using a Lira reagent (Biolabmix, Novosibirsk, Russia). Isolated RNA was treated with 1 μL of DNaseI (Thermo Fisher Scientific, Waltham, MA, USA). Concentration and quality of RNA extracts were determined spectrophotometrically at A280/A260 wavelength absorption ratio on BioSpectrometer (Eppendorf, Hamburg, Germany). cDNA was isolated with oligo(dt) primers, MMLV RT kit (Evrogen, Moscow, Russia), and RNA in equal concentrations. In total, 1 μL of cDNA was used for further reactions.

### 4.10. Real-Time PCR

Specific primers (Table 2) were developed with the use of Primer-BLAST [104]. Since *UaMC2* and *UaALDH12* genes contain introns in their structure, the primers were adjusted to sites at the boundary of two exons, allowing amplification of a fragment using only cDNA. Primer dimer analysis was performed with OligoAnalyzer Tool from ITD [105]. Next, 25 μL of premix with a SYBR Green I intercalating dye. Five qPCRmix-HS SYBR (Evrogen, Moscow, Russia) dyes were used for reaction, according to manufacturer’s recommendations. A reaction mix without cDNA was used as a negative control. The relative gene expression was assessed via reference gene valuation followed with the delta-delta Ct method [85]. Reference gene candidates were selected by analyzing the literature [106,107] and using RefFinder software (available at https://blooge.cn/RefFinder) [108]. Two genes having the most stable expression were chosen: 18S rRNA—the most stable under algicidal exposure and prolonged culture—and E2 (ubiquitin-conjugating enzyme)—the most stable under silicon starvation.

### 4.11. Statistical Analysis

Statistical analysis was performed using the Mann–Whitney U test with the Shapiro–Wilk test of normality in Statistica 7.0. software (Stat Soft Inc., Tulsa, OK, USA). Statistically significant differences were calculated at *p* ≤ 0.05.

## Figures and Tables

**Figure 1 ijms-25-02314-f001:**
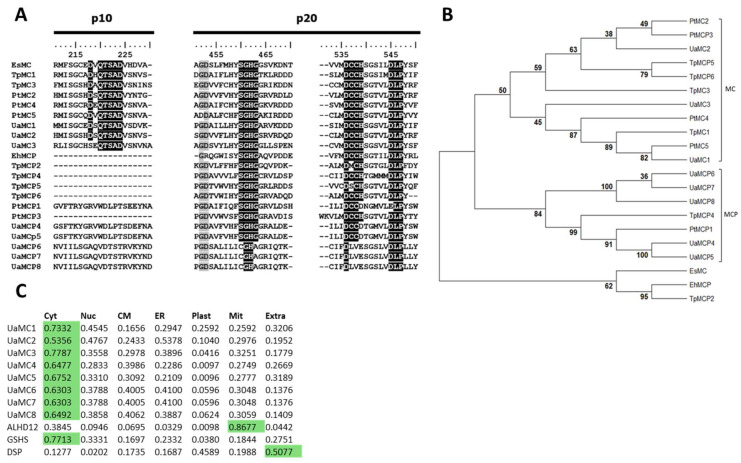
Type III metacaspases and metacaspase-like proteases. (**A**) Partial alignment of UaMC amino acid sequences with metacaspases (MC) and metacaspase-like proteases (MCP) of diatoms *T. pseudonana* (Tp) and *P. tricornutum* (Pt), brown algae *E. siliculosus* (Es), and coccolithophore *E. huxley* (Eh) showing significant amino acid similarity, especially in the conserved, histidine–cysteine catalytic dyads and Ca^2+^ binding site (black shading). Sequences were aligned using ClustalW; grey shading points at conserved amino acid residues. (**B**) Phylogenetic tree built according to alignment of the predicted amino acid sequences of metacaspases (MC) and metacaspase-like proteases (MCP) of diatoms *T. pseudonana* and *P. tricornutum*, brown algae *E. siliculosus*, and coccolithophore *E. huxleyi* obtained using maximum likelihood estimation method (ML) with 1000 bootstrap replications. (**C**) Predicted localization of proteins obtained with the use of DeepLoc 2.0 algorithm; green shading points at high significance assessment. Legend: Cyt, cytosol; Nuc, nucleus; CM, cytoplasmatic membrane; ER, endoplasmic reticulum; Mit, mitochondrion; Extra, extracellular.

**Figure 2 ijms-25-02314-f002:**
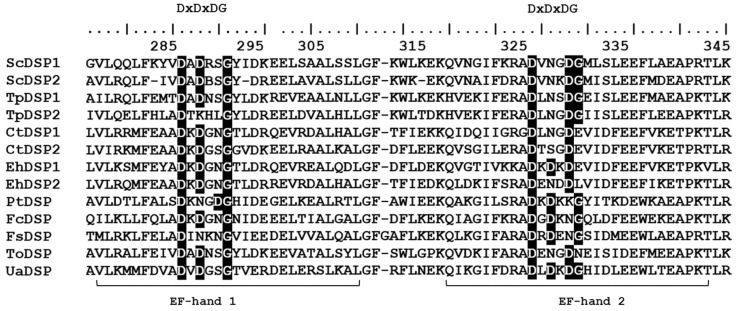
Alignment of the predicted UaDSP amino acid sequences to DSP of other diatoms and coccolithophores (Appendix A). Numbers above point at amino acid positions in conformity with ScDSP1. Black shading emphasizes conserved amino acid residues typical of DxDxDG motif of calcium-binding EF-hand domains. Full alignment is present in Appendix A.

**Figure 3 ijms-25-02314-f003:**
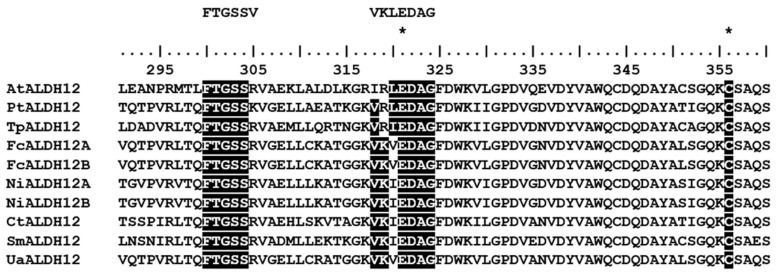
Alignment of the predicted UaALDH12 amino acid sequences to ALDH12 of diatoms and a model organism *A. thaliana*. White letter on black background points at NAD^+^-binding FTGSSV site and VKLEDAG catalytic site. * indicates glutamic acid active site (E) and cysteine acid active site (C).

**Figure 4 ijms-25-02314-f004:**
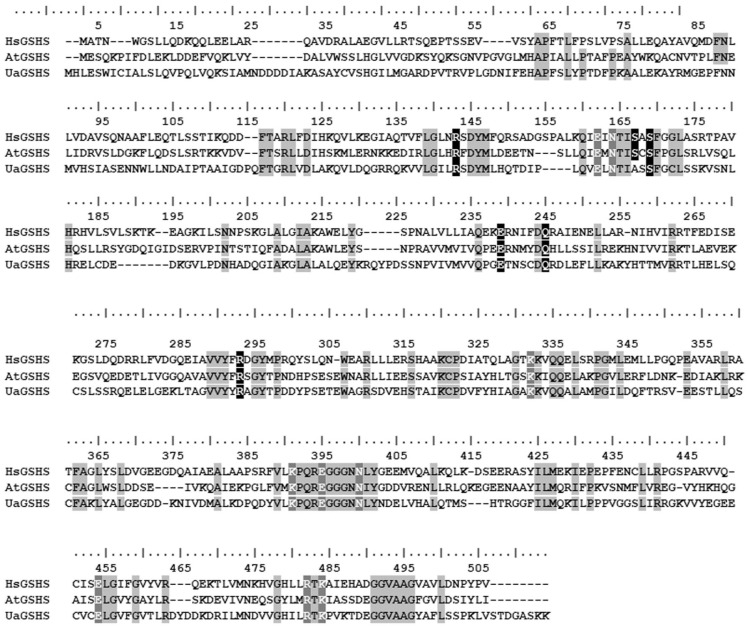
Sequence alignment of the GSHS from *H. sapiens* (HsGSHS, GenBank accession number NP000169.1), *A. thaliana* (AtGSHS, GenBank accession number AAA64781.1), and *U. acus* (UaGSHS). Conserved residues are marked by grey shading. The binding site of γ-glutamylcysteine/glutathione is indicated by white letters on a black background and the ATP-binding site is indicated by white letters on a grey background.

**Figure 5 ijms-25-02314-f005:**
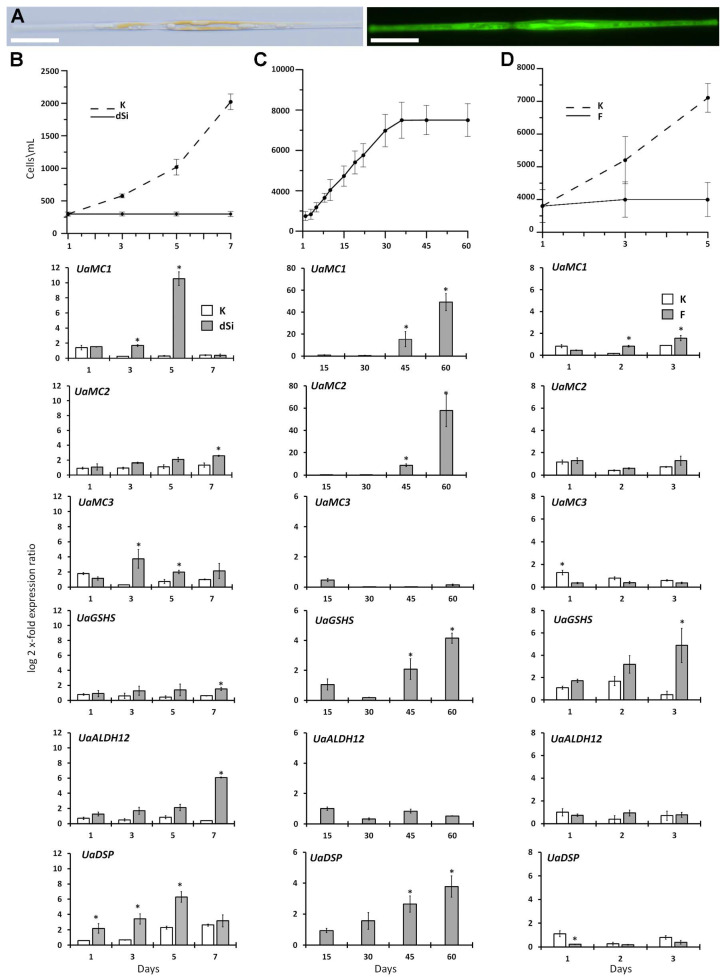
*U. acus* culture growth and relative expression levels of the target genes under various stresses. (**A**) Light and fluorescence microscopy of live *U. acus* cells stained with live cell labeling kit (green), 20 μm scale; (**B**) *U. acus* culture growth curve and differential expression of target genes under silica limitation (K, control; dSi—silica limitation); (**C**) *U. acus* culture growth curve and differential expression of target genes during prolonged culturing; (**D**) *U. acus* culture growth curve and differential expression of target genes under *B. mycoides* filtrate treatment (K—control; F—filtrate). For all values, average and standard deviation of three replicates are shown. * statistically significant difference from control (*p* < 0.05 according to Mann–Whitney U test).

**Table 1 ijms-25-02314-t001:** Type III metacaspase and metacaspase-like protease domain structure.

**Species**	**NCBI Accession ** **Number**	**Type of ** **Metacaspase**	**Length, a.a.**	**Abbreviation**	**p10 ** **Domain**	**p20 ** **Domain**
*Phaeodactylum* *tricornutum*	XP_002178108.1	Metacaspase-like protease	633	PtMCP1		285–588
	XP_002182518.1	Metacaspase type III	369	PtMC2	16–161	174–358
	XP_002180075.1	Metacaspase-like protease	292	PtMCP3		31–257
	XP_002182552.1	Metacaspase type III	322	PtMC4	7–102	116–319
	B7G6D3.2	Metacaspase type III	337	PtMC5	6–111	124–325
*Thalassiosira* *pseudonana*	XP_002294738.1	Metacaspase type III	318	TpMC1	2–112	126–315
	XP_002292692.1	Metacaspase-like protease	208	TpMCP2		20–208
	XP_002287857.1	Metacaspase type III	500	TpMC3	38–273	290–484
	XP_002290139.1	Metacaspase-like protease	347	TpMCP4		55–304
	XP_002297230.1	Metacaspase-like protease	235	TpMCP5		1–204
	XP_002295354.1	Metacaspase-like protease	159	TpMCP6		11–159
*Ulnaria acus*	OR677812	Metacaspase type III	318	UaMC1	7–114	127–304
	OR677813	Metacaspase type III	306	UaMC2	3–103	117–291
	OR677814	Metacaspase type III	426	UaMC3	7–127	141–377
	OR677815	Metacaspase-like protease	701	UaMCP4		372–669
	OR677816	Metacaspase-like protease	701	UaMCP5		372–669
	OR677817	Metacaspase-like protease	320	UaMCP6		113–292
	OR677818	Metacaspase-like protease	320	UaMCP7		113–292
	OR677819	Metacaspase-like protease	320	UaMCP8		113–292
Out group						
*Ectocarpus siliculosus*	CBN76943.1	Metacaspase type III	354	ESMC	42–150	166–353
*Emiliania huxleyi*	XP_005790400.1	Metacaspase-like protease	388	EhMCP		25–263

**Table 2 ijms-25-02314-t002:** Primer sequences for real-time PCR.

**Gene**	**Forward Primer (5′→3** **′)**	**Reverse Primer (5′→3** **′)**
18S rRNA	CTTCTTAGAGGGACGTGCGTTC	TCTCGGCCAAGGTACACTCG
*E2*	AGGATCAGTGGAGTCCTGCT	TTTCGACGTCCACTCTCGTG
*MC1*	TGAGCCAGGTGATGCAATCC	CTTCAGCGGCTTGATGATCGT
*MC2*	CTGGACATGGAAGTCGTGTACG	CTTCGCAAAGGCTTCACAAGC
*MC3*	ACCAATTTCCCGAGCAGGAT	CCATGACCGGAATAGTGCAGA
*DSP*	TGATGGAAGTGGTACGGTGG	GGAGCTTCGGTCAACCACTC
*ALDH12*	GTCGGTCATAGATGGCCTTACG	GAAGATCGACATCGGCAGGA
*GSHS*	ACCCGTAGGTGGCTCATTG	CGTCTTCACTGGCTTTGTGC

## Data Availability

All the necessary links to sites and sequence numbers are contained in the Materials and Methods section of this paper and in the Appendix A.

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
