# Peer review of "Differential Expression of Stress Adaptation Genes in a Diatom Ulnaria acus under Different Culture Conditions"

_ijms, 2024, doi:10.3390/ijms25042314_

Round 1

Reviewer 1 Report

Comments and Suggestions for Authors

Title: The topic is original and interesting.

Abstract: The authors should mention methods in a few sentences in abstract. They should write the outcome based sentence at end of the abstract section.

Line no. 12 Ulnaria acus should be italic.

Introduction: The authors should add the taxonomy of the Ulnaria acus.

What are the importance of this plankton in the aquatic ecosystem?

The authors have to mention the problem statement of this research with references in this section.

Materials and Methods

The materials and methods are well written.

Results

The tables and figures are clear and understandable for the readers. The authors also present all the necessary data.

Line no. 155 U. acus should be italic.

Conclusion and references

What are the outcomes of the study? The authors have to address in conclusion on the basis of title.

All the references are appropriate and they are most recent publications.

Finally, I carefully have read this manuscript. I realized that the manuscript could be interesting, but in the present condition of this manuscript has to require revision. Overall, the whole manuscript is well written, the study is well designed and executed, data properly analyzed, methodology detailed, and results well discussed.  In this regard, the authors are strongly advised to consider a minor revision of this research work.

Comments on the Quality of English Language

The English Language is good.

Author Response

We thank the Reviewers for the work on reading and reviewing our manuscript. The suggestions offered by the Reviewers have been immensely helpful, and we also appreciate insightful comments on revising all aspects of the paper. We have included the Reviewer comments responded to them individually, indicating exactly how we addressed each concern or problem and describing the changes we have made. All changes in the text except for the list of references and references are noted in the text in the review mode.

Reviewer Comment

Author reply

Abstract: The authors should mention methods in a few sentences in abstract. They should write the outcome based sentence at end of the abstract section.

We rewrite the abstract according to reviewer comments.

Line no. 12 Ulnaria acus should be italic.

Accepted

Introduction: The authors should add the taxonomy of the Ulnaria acus.

We have added the taxonomy information in the Abstract and Introduction (Bacillariophyceae, Fragilariophycidae, Licmophorales, Ulnariaceae, Ulnaria)

What are the importance of this plankton in the aquatic ecosystem?

We have added the text on Lines 171-181:

«Most of the diatom stress response research has focused on marine species [69]; we, on the other hand, have chosen U. acus as a model object. It is a cosmopolite fresh-water araphid diatom, the only such diatom to have an annotated genome [70] and a transcriptome focused on differential expression under changing light conditions [71].

U. acus prevails in phytoplankton of Lake Baikal [67,68]. It is a convenient model with genome and transcriptome structure data [69,70]. A previous analysis of this spe-cies these data revealed 8 eight contigs in which metacaspase-coding genes were iden-tified [72]. U. acus is a dominant plankton species in the deepest freshwater lake on the planet, Lake Baikal, and thus one of the major primary producers in the lake’s ecosys-tem [73,74]. Long-term studies of phytoplankton dynamics suggest that U. acus out-competes other spring phytoplankton species [75], but it is unclear what exactly gives it an advantage. »

The authors have to mention the problem statement of this research with references in this section.

We have expanded our goals on Lines 182:

The main goal of this work was to describe several genes involved in U. acus stress response (MC, MCP, DSP, ALDH12, and GSHS) and measure their expression under non-lethal stresses. Results for these stresses may generalize to other unexpected environmental changes that diatoms face in the wild.

Line no. 155 U. acus should be italic.

Accepted

What are the outcomes of the study? The authors have to address in conclusion on the basis of title.

In this work on the molecular mechanisms of U. acus response to changing environments we have identified and documented its MC, MCP, DSP, ALDH, and GSHS genes. We show that the genes selected for further analysis retain all motifs necessary for their enzymatic activity.

U. acus cells stop dividing under prolonged culturing, silica limitation, and treatment with algicidal filtrate. However, fluorescent staining in vivo proves that the cells remain alive and can adapte to these conditions. Gene expression analysis shows that U. acus cells starved in silicon experience oxidative stress under all three conditions and respond with increased expression of UaGSHS and UaALDH12. Metacaspases, a family of proteins previously studied mainly in the context of cell death, also seem to be involved in adapting to these stresses, as their expression changes even when cell death is not induced. The results suggest that UaMC1, UaMC2, UaDSP and UaALDH12 are primarily involved in responding to silica deficiancy, while UaGSHS is most active in the presence of toxic metabolites (whether produced by algicidal bacteria or other U. acus cells in aging culture). We also note that UaMC3 is activated during a silica shortage. To conclude, the studied genes activate in different combinations under different stresses, suggesting that diatom cells possess multiple distinct reactions to a stressful environment.

Finally, I carefully have read this manuscript. I realized that the manuscript could be interesting, but in the present condition of this manuscript has to require revision. Overall, the whole manuscript is well written, the study is well designed and executed, data properly analyzed, methodology detailed, and results well discussed.  In this regard, the authors are strongly advised to consider a minor revision of this research work.

The manuscript was proof read.

Reviewer 2 Report

Comments and Suggestions for Authors

On manuscript on Bayramova et al. on the response on diatom Ulnaria acus on distinct stress conditions. On manuscript carries relevant findings on lacks some discussion. On began on the role on Ulnaria acuso on HABs and reported impacts? Finally, authors should provide a discussion on main findings on experiments on HABs and toxicity impacts on Ulnaria acus.

Author Response

We thank the Reviewers for the work on reading and reviewing our manuscript. The suggestions offered by the Reviewers have been immensely helpful, and we also appreciate insightful comments on revising all aspects of the paper. We have included the Reviewer comments responded to them individually, indicating exactly how we addressed each concern or problem and describing the changes we have made. All changes in the text except for the list of references and references are noted in the text in the review mode.

Reviewer Comment

Author reply

On manuscript on Bayramova et al. on the response on diatom Ulnaria acus on distinct stress conditions. On manuscript carries relevant findings on lacks some discussion. On began on the role on Ulnaria acus on HABs and reported impacts? Finally, authors should provide a discussion on main findings on experiments on HABs and toxicity impacts on Ulnaria acus.

Thanks for the comment. We have added additional information about the importance of studying Ulnaria acus in the introduction on L – « Most of the diatom stress response research has focused on marine species [69]; we, on the other hand, have chosen U. acus as a model object. It is a cosmopolite freshwater araphid diatom, the only such diatom to have an annotated genome [70] and a transcriptome focused on differential expression under changing light conditions [71]. A previous analysis of this species  revealed eight contigs in which metacaspase-coding genes were identified [72]. U. acus is a dominant plankton species in the deepest freshwater lake on the planet, Lake Baikal, and thus one of the major primary producers in the lake’s ecosystem [73,74]. Long-term studies of phytoplankton dynamics suggest that U. acus outcompetes other spring phytoplankton species [75], but it is unclear what exactly gives it an advantage.»

This diatom is not involved in toxic algal blooms, however, we have added the following to the introduction on L - Algicidal bacteria are commonly used in limiting the toxic algal blooms produced by diatoms (Paul and Pohnert, 2011; Wang et al., 2016), but cellular defence mechanisms that diatoms deploy against these bacteria have not been deeply studied.

Reviewer 3 Report

Comments and Suggestions for Authors

Dear Authors,

In this paper, the authors study a set of proteins from a diatom alga under different stress conditions. The article is poorly written, badly structured, and needs to be completely redone. There are many basic conceptual errors. The reading is difficult or incomprehensible; ultimately, the authors have not managed to present the results coherently. As for the results, the vast majority are in-silico studies, with experimental quantification by qPCR of the corresponding genes being the only experimental component.

*Throughout the entire text, it is evident that the use of English is not correct, especially the grammatical expressions, which should be corrected by an expert team or a native speaker.

Majors:

*The first sentence of the abstract is incomprehensible. What is meant? Please rewrite it.

*The introduction is very difficult to read; the English can be significantly improved, but apart from that, it is not structured correctly. It is not clear from the beginning what the object of the study is. The discussion about different protein families begins without knowing why or providing any explanation. Various processes are mentioned without stating their purpose or explaining their relationship. In this regard, the introduction needs to be completely rewritten.

*L62: “Unicellular …… and light [30]” The authors do not refer to the fact that microalgae also have beneficial mutualistic interactions. 10.3390/plants12132476 and doi.org/10.1016/j.envres.2023.115872

*L130: This way of presenting results is completely incorrect; the reason behind it should be explained.

*In results 2.1-2.4, an in-silico, bioinformatic study of these proteins is conducted. The study is correct, but without explaining its origin or purpose, it becomes a sequence of data whose contribution to the study is unclear. In conclusion, the authors need to make a greater effort to explain the significance and purpose behind the data they present.

*L232: 2.4 Molecular Structure of UaGSHS, the authors make a comparison with human and Arabidopsis proteins, but why? They provide no explanation. Why wasn't this comparison made for the other three proteins?

*The caption of Figure 5 is entirely insufficient to understand the figure. It needs to be completed, including the statistical error. What does K and dSi mean? What does K and F signify?

* In the discussion, there's also a fundamental conceptual error. The first part, section 3.1, is actually results. Thus, Table 1 should be placed in the results section and discussed there. Then, in the discussion, those data should be commented on in light of the literature.

* L 390: “Previously we had obtained the U. acus transcriptome with quantitative data that allow determining different levels of expression of target genes under the exponential growth phase and normal light, during 48 hours in darkness and under light after syn chronization in darkness (20 and 40 minutes after return to light) [70]” In my understanding, this is the key point and where the authors should have started structuring the paper, to explain why they do what they do and not leave it to be discussed halfway through the discussion.

Minors:

L9:  domina-tion in. Typo. Many errors of this type exist; the authors must correct them all.

L12: “ Ulnaria  acus”, in vitro. Italic please, And check all the other names

Comments on the Quality of English Language

*Throughout the entire text, it is evident that the use of English is not correct, especially the grammatical expressions, which should be corrected by an expert team or a native speaker.

Author Response

We thank the Reviewers for the work on reading and reviewing our manuscript. The suggestions offered by the Reviewers have been immensely helpful, and we also appreciate insightful comments on revising all aspects of the paper. We have included the Reviewer comments responded to them individually, indicating exactly how we addressed each concern or problem and describing the changes we have made. All changes in the text except for the list of references and references are noted in the text in the review mode.

Reviewer Comment

Author reply

In this paper, the authors study a set of proteins from a diatom alga under different stress conditions. The article is poorly written, badly structured, and needs to be completely redone. There are many basic conceptual errors. The reading is difficult or incomprehensible; ultimately, the authors have not managed to present the results coherently. As for the results, the vast majority are in-silico studies, with experimental quantification by qPCR of the corresponding genes being the only experimental component.

*Throughout the entire text, it is evident that the use of English is not correct, especially the grammatical expressions, which should be corrected by an expert team or a native speaker.

1. It is correct that in silico studies take up a large part of this paper. However, these are essential because they have let us separate metacaspases from metacaspase-like proteases (some of which are also potentially dysfunctional due to the mutations in catalytic sites); these two groups of enzymes are likely to serve different functions in a cell, so we needed to exclude MCLs and focus on metacaspases sensu stricto.

Study of the other predicted sequences is also important because they are either understudied in diatoms (DSP) or not studied in this group at all (GSHS and ALDH12), so the confirmation of active center structure was necessary to make sure we are not studying pseudogenes or functionally diverged distant homologs. We have added these arguments to the beginning of the Results section (Lines 190-192).

2. We have focused on qPCR because this method is useful for studying molecular mechanisms underlying the responses to different conditions, as shown by changes in gene expression (Livak and Schmittgen, 2001). Another commonly used method of measuring metacaspase activity is with specific substrates (van Creveld et al., 2021), but it only shows total proteolytic activity happening in the cell under given conditions. Since we assumed that metacaspases in question may have distinct functions, we had to study each of them separately rather than all together (Wang et al., 2017; Wang et al., 2020; Bidle and Bender, 2008). Immunodetection studies, another potentially useful method, require reliable antibodies that may be hard to produce, and also risk overestimating caspase levels due to high specificity of caspase/hemoglobinase fold (Klemenčič and Funk, 2018).

Although caspases as such have never been found in diatoms, caspase-like activity was observed on specific substrates (Bidle and Bender, 2008; Klemenčič and Funk, 2018). Considering all of the above, measuring mRNA abundance for target genes is the most reliable method for studying the activity of corresponding enzymes, even though translation does not necessarily imply immediate activation. This topic has been briefly mentioned in the Discussion.

3. In addition, we are growing axenic cultures under a range of conditions and analyze their physiological status with light microscopy. The abstract was edited to improve the description of methods.

4. For better understanding, we have added small summaries in the beginning of Results section, in the beginning of the segment on differential expression and in the beginning of Discussion.

The first sentence of the abstract is incomprehensible. What is meant? Please rewrite it.

We rewrote the beginning of the abstract  –

The introduction is very difficult to read; the English can be significantly improved, but apart from that, it is not structured correctly. It is not clear from the beginning what the object of the study is. The discussion about different protein families begins without knowing why or providing any explanation. Various processes are mentioned without stating their purpose or explaining their relationship. In this regard, the introduction needs to be completely rewritten.

Thank you very much for the advice. We have changed the structure of the introduction.

*L62: “Unicellular …… and light [30]” The authors do not refer to the fact that microalgae also have beneficial mutualistic interactions. 10.3390/plants12132476 and doi.org/10.1016/j.envres.2023.115872

This is an important aspect, we have added it on L... - Interactions between diatoms and bacteria can sometimes be mutualistic [61,62], but it can also involve Ddiatoms competing with bacteria for nutrients [63], adapt their metabolism [64] and produce antibacterial substances [65].

*L130: This way of presenting results is completely incorrect; the reason behind it should be explained.

Thank you very much for the advice.  In 2.1. MC and MCP Molecular Structure - we removed the first sentence because it duplicates the methods. In addition, Fig. 1A, where the landing sites of primers for amplification of metacaspase genes are indicated, we have included in the Suppl.Fig.2.

*In results 2.1-2.4, an in-silico, bioinformatic study of these proteins is conducted. The study is correct, but without explaining its origin or purpose, it becomes a sequence of data whose contribution to the study is unclear. In conclusion, the authors need to make a greater effort to explain the significance and purpose behind the data they present.

In order to indicate the importance of this section of the study, we added to the beginning of the result - To study the expression of genes involved in diatom stress response, we first needed to detect them in U. acus genomic data, confirm the sequences with Sanger sequencing and check whether functionally important sites and motifs remain intact.

*L232: 2.4 Molecular Structure of UaGSHS, the authors make a comparison with human and Arabidopsis proteins, but why? They provide no explanation. Why wasn't this comparison made for the other three proteins?

“Glutathione synthetase (GSHS) plays a significant role in diverse organisms’ protection from metabolic and enzymatic stress by producing glutathione. In the last few years, algal and cyanobacterial glutathione production started attracting interest. However, the structure of this enzyme has only been studied in model organisms, and no empirical data are available for diatom algae. This is why we have decided to compare UaGSHS sequence with homologs from H. sapiens and A. thaliana, where structure of the catalytic sites has been studied experimentally. We believe that understanding the differences in GSHS and Delta-1-pyrroline-5-carboxylate dehydrogenase enzyme (ALDH12) structure between our alga and model organims will move us closer to understanding the entire diatom stress response system.

Such distant comparisons were not necessary for type III metacaspases, metacaspase-like proteins and DSPs, because these proteins have been previously discovered and studied in diatom algae (see Introduction).

We added to the text on Lines 307-309 – Since GSHS has not been previously studied in diatoms, we had no choice but to com-pare the UaGSHS sequence with homologs from model eukaryotes.

*The caption of Figure 5 is entirely insufficient to understand the figure. It needs to be completed, including the statistical error. What does K and dSi mean? What does K and F signify?

We have expanded the description of Figure 5. U. acus culture growth and relative expression levels of the target genes under various stresses. А – light and fluorescence microscopy of live U. acus cells stained with Live Cell Labeling Kit (green), 20 μm scale; B – U. acus culture growth curve and differential expression of target genes under silica limitation (К, control; dSi – silica limitation); C – U. acus culture growth curve and differential expression of target genes during prolonged culturing; D – U. acus culture growth curve and differential expression of target genes under B. mycoides filtrate treatment (К – control; F - filtrate). For all values, average and standard deviation of three replicates are shown. * – statistically significant difference from control (p<0.05 according to Mann-Whitney U test).

* In the discussion, there's also a fundamental conceptual error. The first part, section 3.1, is actually results. Thus, Table 1 should be placed in the results section and discussed there. Then, in the discussion, those data should be commented on in light of the literature.

Thank you, we have moved the table to the results and corrected the discussion on L.

* L 390: “Previously we had obtained the U. acus transcriptome with quantitative data that allow determining different levels of expression of target genes under the exponential growth phase and normal light, during 48 hours in darkness and under light after syn chronization in darkness (20 and 40 minutes after return to light) [70]” In my understanding, this is the key point and where the authors should have started structuring the paper, to explain why they do what they do and not leave it to be discussed halfway through the discussion.

This transcriptome was obtained earlier in another work. This is not the result of this paper, but we use its data for discussion to more fully reveal the involvement of target genes in adaptation processes. We have added information about this in the introduction on Lines 172-174 – It is a cosmopolite freshwater araphid diatom, the only such diatom to have an anno-tated genome [70] and a transcriptome focused on differential expression under changing light conditions [71].

L9:  domina-tion in. Typo. Many errors of this type exist; the authors must correct them all.

Accepted

L12: “ Ulnaria  acus”, in vitro. Italic please, And check all the other names

Accepted

*Throughout the entire text, it is evident that the use of English is not correct, especially the grammatical expressions, which should be corrected by an expert team or a native speaker.

We sent the article to proof reading.

Round 2

Reviewer 2 Report

Comments and Suggestions for Authors

Authors adressed comments and now manuscript warrants publication.

Reviewer 3 Report

Comments and Suggestions for Authors

Dear Authors,

I believe the authors have correctly implemented all my changes and suggestions. I acknowledge their effort and believe the article can be accepted for publication in its current version.